# Cadmium Tolerance and Detoxification Mechanisms of *Lentinula edodes*: Physiology, Subcellular Distribution, and Chemical Forms

**DOI:** 10.3390/microorganisms13010062

**Published:** 2025-01-02

**Authors:** Gaigai Chen, Bowen Han, Wene Nan, Xiaobo Dong

**Affiliations:** 1College of Food Science and Engineering, Northwest A&F University, Yangling 712100, China; 2Edible Fungi Center, Northwest A&F University, Yangling 712100, China

**Keywords:** cadmium, *Lentinula edodes*, detoxification, subcellular distribution, chemical forms

## Abstract

*Lentinula edodes* has a strong cadmium-enrichment ability, posing a potential threat to human health. However, the cadmium tolerance and detoxification mechanisms of *Lentinula edodes* are not understood. We investigated the physiological responses, subcellular distribution, and chemical forms of cadmium in two *Lentinula edodes* strains (1504 and L130) with contrasting cadmium tolerance. The results showed that appropriate, low-level cadmium promoted mycelial growth, and higher cadmium exposure induced obvious inhibition of mycelial growth by damaging the cell wall and membrane structure and triggering the overproduction of ROS. Antioxidant enzymes played an important role in cadmium detoxification, as well as functional group modulation. Cadmium was predominantly distributed in the cell wall fraction, and NaCl-extractable cadmium was the main chemical form. Enhanced antioxidant enzyme activities, reduced cadmium accumulation, and increased HAc-extractable cadmium with less toxicity promoted stronger cadmium tolerance and detoxification abilities in L130 compared to 1504. Thus, this study provides new insights into cadmium tolerance and detoxification in *Lentinula edodes*.

## 1. Introduction

*Lentinula edodes* is the second-most cultivated edible mushroom in the world. Owing to its high nutritional and medicinal value and unique flavor, *Lentinula edodes* is popular worldwide, particularly in some Asian countries. China is the largest producer of *Lentinula edodes* in the world, accounting for over 95% of the world’s production [1].

Cadmium is a non-essential heavy metal with high toxicity and mobility [2]. Cadmium has been reported to be the most serious heavy metal pollutant in China [3]. Cadmium, a group 1 carcinogen, can enter the human body via the food chain, causing itai-itai disease, cardiovascular disease, and liver dysfunction [4,5]. It has been reported that some macrofungi can accumulate cadmium with a cadmium accumulation capacity much higher than that of plants. *Pleurotus ostreatus* grown in cadmium-polluted soil can accumulate 15.6 mg/kg cadmium, and the accumulation of cadmium in mycelia cultured in a cadmium-contaminated medium can reach 340 mg/kg [6,7]. *Cystoderma carcharias*, a cadmium hyperaccumulator, can accumulate 604 mg/kg cadmium (dry weight) in smelter-polluted areas in fruiting bodies [8]. In *Lentinula edodes*, the mycelia can accumulate 141.6 mg/kg cadmium after 24 h of cultivation with 50 μM cadmium [9]. The high cadmium accumulation in macrofungi poses potential threats to human health. The notable enrichment of cadmium in *Lentinula edodes* means that it has an exceptional ability to perform cadmium detoxification.

Fungi have evolved complex mechanisms against cadmium stress, including antioxidant defense systems, extracellular fixation, and intracellular compartmentalization [10]. Cadmium can induce the overproduction of reactive oxygen species (ROS), disrupting cellular redox balance and normal metabolism [11]. Antioxidant enzymes, such as superoxide dismutase (SOD), catalase (CAT), and peroxidase (POD), can be activated by ROS and convert superoxide radicals to H_2_O and oxygen through a series of reactions [2]. It has been reported that fungi can scavenge excess ROS by enhancing the activity of antioxidant enzymes and repairing oxidative damage, and the variation in the activity of antioxidant enzymes under cadmium stress depends highly on the fungal species [11]. The cell wall is the first barrier against cadmium. A variety of functional groups on the cell wall, such as hydroxyl and carboxylic groups, can interact with cadmium, facilitating cadmium accumulation and detoxification in fungi [12]. Then, cadmium can enter cells via certain essential metal ion transportation channels and be partially compartmentalized in vacuoles. The subcellular distribution and chemical forms of cadmium can affect the accumulation of and tolerance to cadmium in organisms. Cadmium in the organelle fraction shows higher toxicity than in the cell wall and vacuoles [13]. It was found that in *Lemna minor*, *Porphyra yezoensis*, and *Agrocybe aegerita*, cadmium was mainly distributed in the cell wall or vacuole fractions [11,14,15]. This suggested that the precipitation of cadmium in the cell wall and its sequestration in vacuoles directly participated in the detoxification of cadmium in organisms. The chemical forms of cadmium are directly associated with its migration capacity and toxicity. In general, water-soluble cadmium in organic and inorganic forms exhibits higher toxicity than insoluble cadmium in pectate and integrated pectate–phosphate forms [14]. Studies have revealed that the dominant chemical form of cadmium is species-dependent. In *Porphyra yezoensis*, the greatest quantity of cadmium was precipitated with phosphate under high cadmium stress, whereas in *Siegesbeckia orientalis*, pectate-, oxalate-, and protein-integrated cadmium were the predominant forms [15,16]. Various *Lentinula edodes* strains exhibit significantly different capacities for cadmium tolerance and detoxification. However, the cadmium tolerance and detoxification mechanisms of *Lentinula edodes* have been unclear.

Thus, two *Lentinula edodes* strains with contrasting cadmium tolerance were selected in the present study. The aim of this study was to investigate the physiological response of *Lentinula edodes* to different concentrations of cadmium, as well as the subcellular distribution and chemical forms of cadmium. This study will provide valuable insights into the cadmium tolerance and detoxification mechanisms of *Lentinula edodes*.

## 2. Materials and Methods

### 2.1. Materials and Cultivation of L. edodes Mycelia

Malt extract, yeast extract, peptone, agar, and Tris base were purchased from Solarbio (Beijing, China). Trichloroacetic acid, glutaraldehyde, CdCl_2_, HNO_3_, HCIO_4_, and other chemicals were all analytical reagents from Sinopharm (Beijing, China).

*L. edodes* strains (L130 and 1504) were provided by the Wuhan Academy of Agricultural Sciences, Wuhan, China. The mycelia were cultivated in MYG solid medium (2% glucose, 2% malt extract, 0.1% yeast extract, 0.1% peptone, and 2% agar) containing 0–100 μmol/L Cd(II) at 25 °C.

### 2.2. Determination of Mycelial Growth Rate and Biomass

The mycelial growth rate was measured by the cross-over method [17]. The growth rate of mycelia was calculated with vernier-based measurements of the colony diameter, determined 7 days after inoculation. The growth rate was calculated using the following equation:The growth rate (mm/d) = (D − 7.5)/7(1)

D is the diameter of the colony diameter; 7.5 is the diameter of the inoculum.

The harvested mycelia were used to determine the fresh biomass immediately, and the dry biomass was measured after the mycelia were dried to a constant weight.

### 2.3. Scanning Electron Microscopy (SEM)

The mycelial morphology was observed using SEM according to Wu et al. [18]. The collected mycelia were washed with a PBS solution and fixed with 2.5% glutaraldehyde at 4 °C for 24 h. The fixed mycelia were dehydrated with graded ethanol and dried by a critical point dryer. Then, the resulting samples were coated with gold for SEM observation (Nano SEM-450, FEI, Waltham, MA, USA).

### 2.4. Transmission Electron Microscopy (TEM)

The cellular microstructure of the mycelium was visualized using TEM according to Wu et al. [19], with minor modifications. The mycelium was fixed using 2.5% glutaraldehyde at 4 °C for 10 h, dehydrated with graded ethanol, and then embedded in epoxy resin. Ultrathin sections sliced with an ultrathin microtome (EMUC7, Leica, Wetzlar, Germany) were double-stained with uranyl acetate and lead citrate. Then, the resulting samples were embedded in a copper grid for TEM observation (TECNAI G2 SPIRIT BIO, FEI, USA).

### 2.5. Fourier Transform Infrared Spectroscopy (FTIR)

FTIR was performed using a Vertex7 spectrometer (Bruker, Karlsruhe, German) according to Tian et al. [20]. The freeze-dried mycelia powder mixed with KBr was ground and then scanned 64 times by FTIR in the range of 4000–400 cm^−1^ at a resolution of 4 cm^−1^.

### 2.6. Determination of Cell Membrane Integrity and Permeability

The cell membrane integrity was monitored by measuring the malondialdehyde (MDA) content according to Meng et al. [21]. The mycelia were mixed with 10% trichloroacetic acid and centrifuged at 4000 rpm for 10 min. Then, the resulting sample was mixed with 0.6% thiobarbituric acid and incubated at 100 °C for 15 min. The cooled supernatant was used to detect OD values at 450 nm, 532 nm, and 600 nm. The MDA concentration was calculated according to the following equation (A represents absorbance):MDA concentration (μmol/L) = 6.45 × (A_532_ − A_600_) − 0.56 × A_450_(2)

The relative electrical conductivity was employed to evaluate the cell membrane permeability according to Liu et al. [22]. The relative electrical conductivity was detected using a conductivity meter (DDS-307A, LEICI, Shanghai, China).

### 2.7. Determination of Antioxidant Enzyme Activity and ROS Contents

Fresh mycelia were harvested and washed with a PBS solution. The activity of antioxidant enzymes, including superoxide dismutase (SOD), catalase (CAT), and peroxidase (POD), and the content of ROS (H_2_O_2_ and O_2_^−^) were determined using Solarbio assay kits (Beijing, China) [23].

### 2.8. Subcellular Distribution of Cadmium in Mycelium

The extraction of cadmium subcellular distribution was performed using the differential centrifugation technique described by Jia et al. [24], with some modifications. The mycelia were homogenized in pre-cooled buffer (250 mM sucrose, 1.0 mM dithioerythritol, 50 mM Tris-HCl with pH 7.5). The homogenate was centrifuged at 300× *g* for 30 min at 4 °C to obtain the cell wall fraction. The resulting supernatant was centrifuged at 20,000× *g* for 45 min at 4 °C. The resultant precipitate and supernatant were defined as the organelle fraction and soluble fraction, respectively. The collected fractions were digested with HNO_3_:HCIO_4_ (4:1) in a microwave digestion instrument (Multiwave PRO, Anton Paar Gmbh, Graz, Austria), and the cadmium concentration was determined using a graphite furnace atomic absorption spectrophotometer (GFAAS, ZA3000, Tokyo, Japan).

### 2.9. Cadmium Chemical Forms

The different chemical forms of cadmium were successively extracted by designated solutions according to Luo et al. [25]. The extraction was conducted in the following order: (1) 80% ethanol for inorganic cadmium; (2) deionized water for water-soluble organic cadmium; (3) 1 M NaCl for pectate- and protein-integrated cadmium; (4) 2% acetic acid for insoluble cadmium–phosphate complexes; (5) 0.6 M HCl for cadmium–oxalate; (6) residual cadmium. The mycelia were homogenized in an extraction solution and shaken at 25 °C for 22 h. Then, the homogenate was centrifuged at 5000× *g* for 10 min to obtain the first supernatant. The precipitate was then re-suspended twice in the extraction solution, shaken at 25 °C for 2 h, and centrifuged. The supernatants collected from three centrifugations were pooled, and the residue was subjected to the next extraction. The resulting five extraction solutions and residue were evaporated on an electric plate and digested with HNO_3_: HCIO_4_ (4:1) to determine the cadmium concentration.

### 2.10. Determination of Cadmium Content

The cadmium content was determined by a GFAAS using cadmium chloride as the standard according to Dong et al. [10]. NH_4_H_2_PO_4_ (0.5%, *w*/*v*) was employed as the matrix modifier. The instrument parameters were as follows: cadmium lamp current of 4.0 mA, wavelength of 228.8 nm, and slit width of 0.5 nm.

### 2.11. Statistical Analysis

All experiments and determinations were conducted in triplicate. The results are presented as the mean ± standard deviation. Significant differences in the mean values (*p* < 0.05) were determined using one-way analysis of variance (ANOVA).

## 3. Results and Discussion

### 3.1. Effects of Cadmium Stress on Mycelia Growth

The morphology, growth rate, and biomass of mycelia were measured to assess the performance of *L. edodes* under different cadmium stress conditions. As shown in Figure 1, under low cadmium stress (<50 μmol/L), the growth rate of both strains was significantly increased in comparison with CK (without cadmium treatment). With cadmium at 10 μmol/L, the fresh weight of 1504 and L130 was 64.84% and 27.70% higher than that of CK, and the dry weight increased by 54.84% and 94.44%, respectively. As expected, the growth rate and biomass of both strains decreased remarkably under higher cadmium stress. For 1504 and L130, the growth rate decreased from 5.82 and 6.48 to 2.41 and 6.24 mm/d, and the fresh weight (dry weight) was reduced from 0.121 (0.019) and 0.167 (0.021) to 0.072 (0.013) and 0.145 g (0.016), respectively. However, when the cadmium concentration was higher than 20 μmol/L, the growth rate and biomass of L130 were significantly higher than those of 1504. With cadmium at 100 μmol/L, the growth rate, fresh weight, and dry weight of L130 were 2.59, 2.10, and 1.23 times higher than those of 1504, respectively. In addition, as shown in Figure 1A, the mycelial diameter of L130 was obviously larger than that of 1504. The results indicated that appropriate, low cadmium stress could promote mycelial growth, while higher cadmium levels would inhibit mycelial growth, and L130 possessed stronger cadmium tolerance. A similar phenomenon was also observed in *Pleurotus cornucopiae* and *Pleurotus ostreatus* [7]. The promotion of mycelial growth might result from the Hormesis effect: low-cadmium treatment could induce a beneficial response and enhance normal function to protect mycelia against subsequent stresses [26]. However, higher cadmium stress could interfere with protein synthesis and enzyme activity and induce autophagy and apoptosis, resulting in growth inhibition [27].

### 3.2. Effects of Cadmium Stress on Mycelial Morphology and Cellular Microstructure

To further investigate cadmium toxicity, the mycelial morphology and cellular microstructure were observed by SEM and TEM, respectively. As shown in Figure 2A, for CK and low-cadmium-treated mycelia, the morphology observed by SEM showed a complete structure and smooth surface, and no remarkable differences were observed among them. However, mycelia treated with high cadmium levels were deformed and fractured, and this phenomenon tended to be more severe with increasing cadmium, resulting in mycelia completely losing their shape at 100 μmol/L cadmium. TEM images revealed that the mycelial cells of CK and those treated with low cadmium levels were intact with a regular shape (Figure 2B). However, cellular ultrastructure showed noticeable deformation under high cadmium stress. It is worth noting that at 100 μmol/L cadmium, for 1504, the cell wall and membrane were severely broken, and obvious cellular leakage occurred, whereas L130 still exhibited an intact cell wall and membrane. These results reveal that low cadmium exposure did not cause alterations in cell structure, while higher cadmium stress could damage the cell structure of mycelia. Compared with 1504, L130 had a more powerful capacity to resist cadmium, protecting the cell from being broken under high cadmium stress. The cell wall’s interception of cadmium and the Hormesis effect might contribute to maintaining the cell integrity and structure under low-cadmium treatment [11,26]. High cadmium stress could induce the rupture of the cell wall and membrane and affect the synthesis and transport of substances, which would disturb normal physiological metabolism and inhibit mycelial growth [28,29].

### 3.3. Effects of Cadmium Stress on Cell Membrane Permeability and Integrity

The cell membrane can maintain the regionalization of intracellular activities. To evaluate the impacts of cadmium stress on cell membrane function, cell membrane permeability and integrity were investigated.

The relative electric conductivity can reflect the permeability of the cell membrane [22]. MDA, as the product of membrane lipid peroxidation, is the key index for evaluating membrane integrity [28]. As shown in Figure 3, the relative electric conductivity tended to increase with increasing cadmium exposure in both strains, and a similar tendency was observed for MDA. Compared with CK, the relative electric conductivity and MDA content showed no significant difference at 10 μmol/L cadmium. However, the relative electric conductivity of 1504 and L130 treated with high cadmium concentrations (50–100 μmol/L) increased by 78.47–152.25% and 30.75–84.27% relative to CK, respectively. Meanwhile, the MDA content in 1504 and L130 increased by 111.99–183.25% and 103.44–183.60%, respectively. Further, it is worth noting that the relative electric conductivity and MDA content in 1504 were significantly higher than those in L130 under high cadmium stress. As expected, the changes in cell membrane permeability were consistent with the changes in integrity. The results suggest that appropriate, low cadmium exposure does not cause alterations in cell membrane permeability and integrity, while higher cadmium stress will damage the cell by impairing membrane permeability and integrity. The changes in cell membrane permeability and integrity further confirm the observations of cellular microstructure by TEM. Similarly, a previous study reported that cadmium stress increased MDA content in common wheat, and the content of MDA in a cadmium-sensitive cultivar (Luomai23) was higher than that in a cadmium-tolerant cultivar (Zhongyu10) [30]. The cell wall is the first barrier in fungi against cadmium stress, and variations in the cell wall will inevitably induce alterations in the cell membrane [19]. The cell wall can protect the membrane from being damaged by chelating a certain amount of cadmium, whereas high cadmium stress exceeding the resistance threshold value of the cell wall will induce the peroxidation of plasma membranes, resulting in the disruption of membrane permeability and integrity [31]. This may explain the alterations in cell membrane permeability and integrity under different cadmium stress levels.

### 3.4. FTIR Analysis/Effects of Cadmium Stress on Functional Groups

FTIR spectra were used to evaluate alterations in the functional groups of cellular components under cadmium stress. As shown in Figure 4, the broad and strong peaks at 3389–3336 cm^−1^ were associated with the stretching vibrations of O-H and N-H [32,33]. The sharp peaks at 1655–1653, 1549–1546, and 1250–1244 cm^−1^ corresponded to amide I (stretching vibration of C=O), II (bending vibration of N-H), and III (stretching vibration of C-N) bands, respectively [10]. The peaks located at 2800–3100 cm^−1^ were due to the antisymmetric vibration of –CH in fatty acids, and the absorption peaks at 900–1200 cm^−1^ represented the characteristic absorptions for polysaccharides [20]. It was observed that the characteristic absorption peaks mentioned above were all enhanced to varying degrees in 1504 and L130 after cadmium treatment. The results suggest that the O-H group of polysaccharides, with a high affinity for divalent cations, might be involved in cadmium bonding. Cadmium stress might induce the dissociation of polymetric O-H groups, contributing to the cadmium tolerance of *L. edodes* mycelia [34]. The carboxylate groups and amino groups in proteins might interact with cadmium and form cadmium–protein complexes [10]. In addition, cadmium stress changed the lipid composition of the cell membrane. Thus, we inferred that cadmium stress induced interactions between functional groups and cadmium, resulting in structural alterations in macromolecular compounds, which ultimately disrupted normal metabolism in *L. edodes* mycelia [35].

### 3.5. Effects of Cadmium Stress on the Antioxidant System

Cadmium stress can stimulate the production of ROS, resulting in cell damage. Fungi have evolved an antioxidant system to resist cadmium-induced oxidative stress, in which antioxidant enzymes play crucial roles, including SOD, CAT, POD, etc. [11]. SOD is the first barrier against ROS and can convert the superoxide anion to molecular oxygen and H_2_O_2_. Then, other antioxidant enzymes (CAT, POD, etc.) act as the second barrier, breaking H_2_O_2_ into H_2_O and O_2_ [2].

As shown in Figure 5A, B, the superoxide anion and H_2_O_2_ tended to increase with increasing cadmium. Under high cadmium stress, the content of the superoxide anion in 1504 was remarkably higher (1.81–2.36 times) than that in L130. The accumulation of ROS induced by cadmium toxicity might exceed the resistance threshold value, resulting in the disruption of the cell membrane, as previously assessed by the MDA level [21]. The SOD and CAT activities across the cadmium treatment concentration gradient showed a low–high–low trend in both strains (Figure 5C,E). The results indicate that cadmium stress stimulated the activities of SOD and CAT, and these two enzymes participated in the detoxification of cadmium. A similar trend was also found in lead-stressed *Trifolium*, as reported by Meng et al. [21]. The decrease in enzyme activities under high cadmium stress might be due to enzyme inactivation or interference with protein synthesis caused by the high levels of ROS induced by cadmium [36]. The POD activity in 1504 showed a low–high–low trend with increasing cadmium; however, L130 exhibited a high–low–high trend, and POD activity was 2.51–34.06 times higher than that in 1504 under high cadmium stress (Figure 5D). The results suggest that, compared with 1504, POD, with higher activity in L130, could better alleviate oxidative stress by scavenging more ROS. This might explain why L130 was more tolerant than 1504 to high cadmium stress. Thus, according to the alterations in MDA, ROS, and mycelial growth and morphology, it can be concluded that antioxidant enzymes play crucial roles in the detoxification of cadmium, and L130 possesses a more powerful antioxidant system, resulting in its stronger cadmium tolerance.

### 3.6. Subcellular Distribution

The cadmium subcellular distributions are shown in Figure 6. The content of cadmium increased from 3.5 to 310.9 mg/kg in 1504 and from 7.5 to 373.7 mg/kg in L130 (Figure 6A,B). The content of cadmium in all subcellular compartments of both strains increased with increasing cadmium exposure. The cadmium content in each subcellular fraction in 1504 was higher than that in L130 (Figure 6C–H). The results suggest that the *L. edodes* mycelia have a strong capacity to enrich cadmium, and the enrichment ability of cadmium-sensitive 1504 is greater than that of the cadmium-tolerant L130 with both low- and high-cadmium treatments. It can be inferred that, compared with the cadmium-sensitive 1504, the cadmium-tolerant L130 can alleviate cadmium toxicity by reducing cadmium accumulation in various subcellular compartments.

The most cadmium was accumulated in the cell wall fractions (61.11–83.26% and 62.14–84.01% in 1504 and L130, respectively), followed by the soluble fractions (14.41–35.16% and 14.39–34.98% in 1504 and L130, respectively), and the organelle fractions accounted for only 2.29–3.73% and 1.60–2.99% in 1504 and L130, respectively (Figure 6I,J). The proportion of cadmium in the cell wall fraction increased with increasing cadmium exposure, while the proportion of cadmium in the soluble and organelle fractions decreased. The results suggest that the compartmentalization of cadmium in the cell wall and soluble fractions plays an important role in elevating the toxicity of cadmium to mycelia. This result is in agreement with Li et al. [11], who reported that cadmium was mainly accumulated in soluble and cell wall fractions in the *Agrocybe aegerita* pileus and stipe.

The cell wall is the first barrier to reduce cadmium toxicity. Functional groups with a negative charge on polysaccharides and proteins of the cell wall, including hydroxyl, carboxyl, and amino groups, could bind to cadmium to restrict cadmium transportation across the cytoplasm [37,38]. Previous studies revealed that most cadmium was distributed in the cell walls of *P. yezoensis* (41.2–79.2%) and *Elodea canadensis* (67%) [15,39]. Thus, the cell wall acted as the predominant sink for cadmium in *L. edodes* mycelia. The increase in the cadmium proportion in the cell wall fraction with increasing cadmium exposure might be explained by the following reasons: (1) more latent binding sites were exposed under higher cadmium stress; (2) the intracellular cadmium-efflux system promoted the efflux of excessive cadmium in the cytoplasm to protect the cell from being damaged [40,41]. Although the cell wall has a great capacity to restrict extracellular cadmium, low-cadmium treatment could still activate transporters in the cell membrane, promoting the entry of cadmium into the cell. The compartmentation of cadmium in vacuoles is a crucial detoxification strategy in various organisms [11]. The cadmium in the soluble fraction was mostly from vacuoles. Organic acids and sulfur-rich peptides in vacuoles, as organo-ligands, could chelate free cadmium ions to decrease their activity, resulting in a reduction in cadmium toxicity [42,43]. It was proven that most cadmium (53–75%) in the pileus and stipe of *A. aegerita* was compartmentalized in the soluble fraction [11]. Thus, we inferred that vacuoles are a key sink for intracellular cadmium in *L. edodes* mycelia, which could alleviate cadmium toxicity and facilitate cadmium accumulation in mycelia.

### 3.7. Chemical Forms of Cadmium

The chemical form of cadmium is closely related to its transportation, accumulation, and toxicity in mycelia. In general, lower polarity means greater migration ability and higher toxicity [14]. The content and proportion of cadmium in different chemical forms distributed in the mycelia are shown in Figure 7. Overall, the cadmium content in different chemical forms showed an increasing trend with increasing cadmium exposure. In *L. edodes* mycelia, the NaCl-extracted state (62.86–75.01% and 44.06–57.85% in 1504 and L130, respectively) was the predominant form of cadmium, followed by H_2_O (17.38–34.44% and 20.90–23.55% in 1504 and L130, respectively), HAc (0.86–11.33% and 18.30–27.35% in 1504 and L130, respectively), ethanol (0.87–1.63% and 1.72–3.90% in 1504 and L130, respectively), and HCl (0.10–0.17% and 0.12–0.25% in 1504 and L130, respectively) extraction states. The order of cadmium toxicity in different chemical forms was F_E_ > F_W_ > F_NaCl_ > F_HAc_ > F_HCl_ [14]. NaCl- and HAc-extractable cadmium mainly formed pectate/protein–cadmium complexes and phosphate–cadmium complexes with less toxicity, respectively. However, cadmium extracted by H_2_O and ethanol was mainly combined with chlorides, nitrate ions, and organic acids, which have higher migration capacity and toxicity [14,44]. Thus, it could be inferred that the *L. edodes* mycelia could convert cadmium from the culture medium into undissolved pectate–, protein–, and phosphate–cadmium complexes rather than soluble inorganic and organic forms to alleviate the toxicity of cadmium to mycelia. This might also explain why the *L. edodes* mycelia possessed a strong cadmium-enrichment ability. A similar result was also found in *Agrocybe aegerita*, where 85–88% cadmium in the fruit body was accumulated in the pileus; the pileus could convert soluble cadmium from the stipe into undissolved cadmium complexes to reduce the toxicity of cadmium to *Agrocybe aegerita*, resulting in higher accumulation in the pileus [11].

In 1504, the proportion of H_2_O- and ethanol-extractable cadmium increased from 18.25% to 36.07% with increasing cadmium exposure, while the proportion in L130 was relatively stable at 22.62–27.45%. In addition, the content of HAc-extractable cadmium in 1504 and L130 showed a decreasing and increasing trend with the increase in cadmium exposure, respectively. Meanwhile, both the content (4.37–11.81 mg/kg) and proportion (18.30–27.35%) of HAc-extractable cadmium in L130 were noticeably higher than those in 1504 (0.40–2.42 mg/kg, 0.86–11.33%). Obviously, L130 accumulated a higher proportion of cadmium with less toxicity than 1504. Thus, the results suggest that, compared with cadmium-sensitive 1504, cadmium-tolerant L130 has a stronger ability to transform cadmium to less toxic forms, which contributes to enhancing its tolerance to high cadmium stress. This result is consistent with that of Wu et al. [45], who reported that a cadmium-sensitive barley genotype accumulated lower pectate- and protein-integrated cadmium than a cadmium-resistant barley genotype.

## 4. Conclusions

In this study, the cadmium stress responses of two different cadmium-tolerant *Lentinula edodes* strains were investigated. The growth and physiological level of mycelia showed a positive stress response to low cadmium exposure. High cadmium stress inhibited growth by damaging the mycelial structure and cell membrane permeability and integrity, particularly for the cadmium-sensitive 1504. Cadmium induced oxidant injuries to mycelia caused by higher levels of ROS, and antioxidant enzymes played an important role in cadmium detoxification. Functional groups (hydroxyl, carboxyl, and amino) were involved in cadmium bonding and detoxification. Cadmium was predominantly distributed in the cell wall fraction, followed by the soluble fraction. NaCl-extractable cadmium was the dominant chemical form. Compared with the cadmium-sensitive 1504, the cadmium-tolerant L130 could better alleviate cadmium toxicity by improving antioxidant enzyme activities, reducing cadmium accumulation in various subcellular compartments, and forming a higher proportion of HAc-extractable cadmium with less toxicity rather than H_2_O- and ethanol-extractable cadmium. To better understand the mechanisms of cadmium accumulation and detoxification in *Lentinula edodes*, additional studies of cadmium transport pathways at the molecular level are warranted.

## Figures and Tables

**Figure 1 microorganisms-13-00062-f001:**
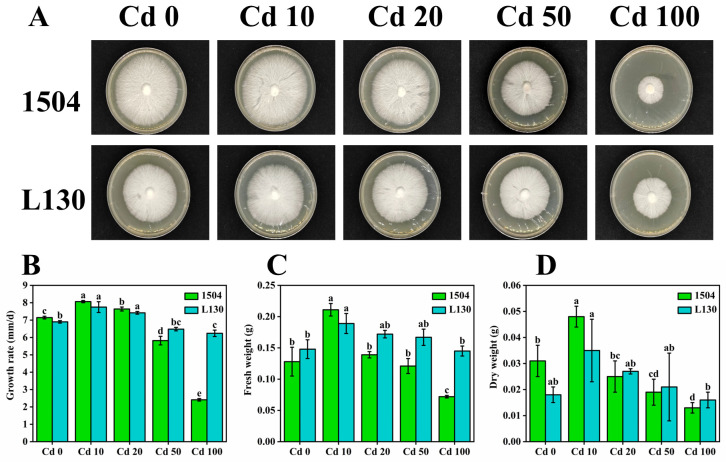
Effects of cadmium stress on mycelia growth. (**A**) Mycelial morphology; (**B**) growth rate; (**C**) fresh weight; (**D**) dry weight. Cd 0, Cd 10, Cd 20, and Cd 50 represent 0, 10, 20, 50, and 100 μM Cd, respectively. Values are presented as mean ± standard deviation (*n* = 3). Small letters indicate significant differences among the varieties and treatments (*p* < 0.05). Values in a column followed by different letters are significantly different (*p* < 0.05).

**Figure 2 microorganisms-13-00062-f002:**
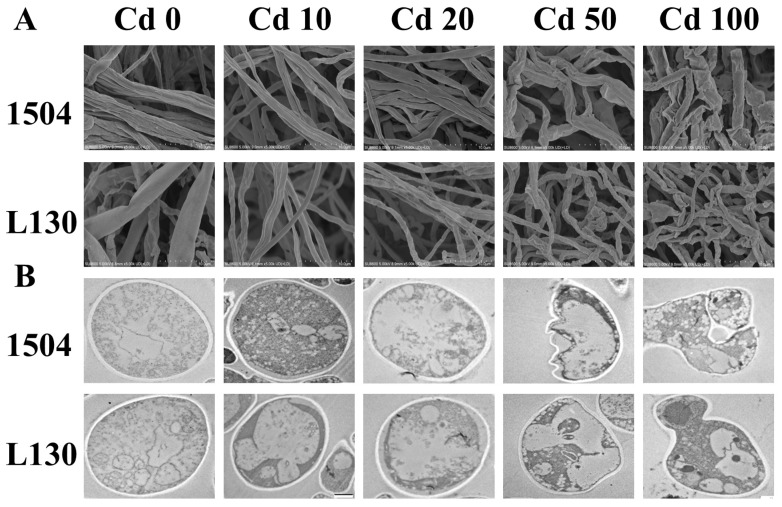
Effects of cadmium stress on mycelial morphology and cellular microstructure. (**A**) Mycelial morphology observed by SEM (5000 × magnification); (**B**) cellular microstructure observed by TEM.

**Figure 3 microorganisms-13-00062-f003:**
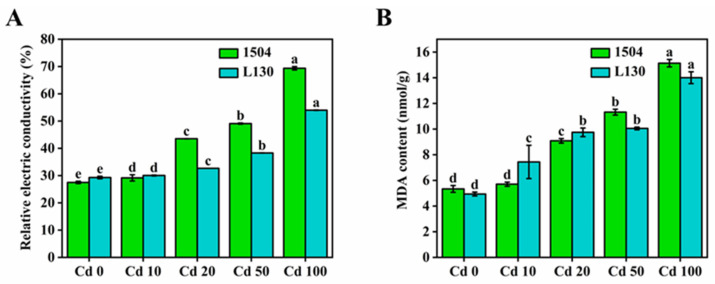
Effects of cadmium stress on cell membrane permeability and integrity. (**A**) Relative electric conductivity; (**B**) MDA content. Values in a column followed by different letters are significantly different (*p* < 0.05).

**Figure 4 microorganisms-13-00062-f004:**
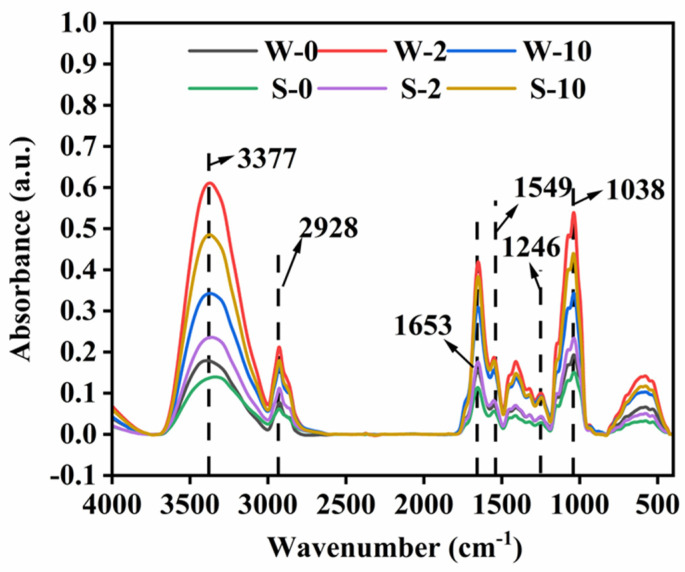
Effects of cadmium stress on functional groups. W means *L. edodes* strain 1504, and S means *L. edodes* strain L130.

**Figure 5 microorganisms-13-00062-f005:**
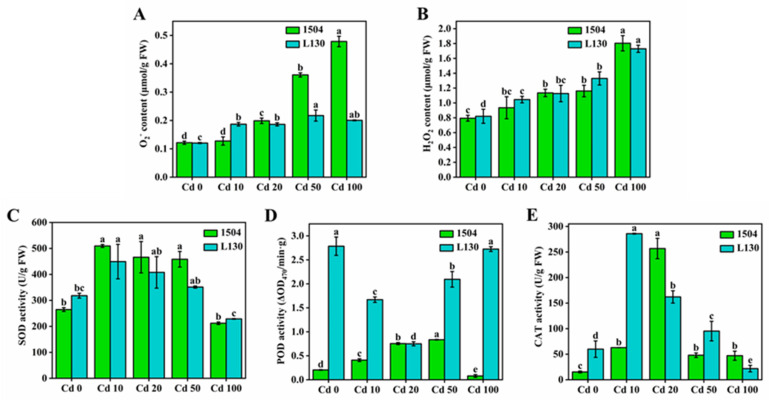
Effects of cadmium stress on the antioxidant system. (**A**) O_2_^−^ content; (**B**) H_2_O_2_ content; (**C**) SOD activity; (**D**) POD activity; (**E**) CAT activity. Values in a column followed by different letters are significantly different (*p* < 0.05).

**Figure 6 microorganisms-13-00062-f006:**
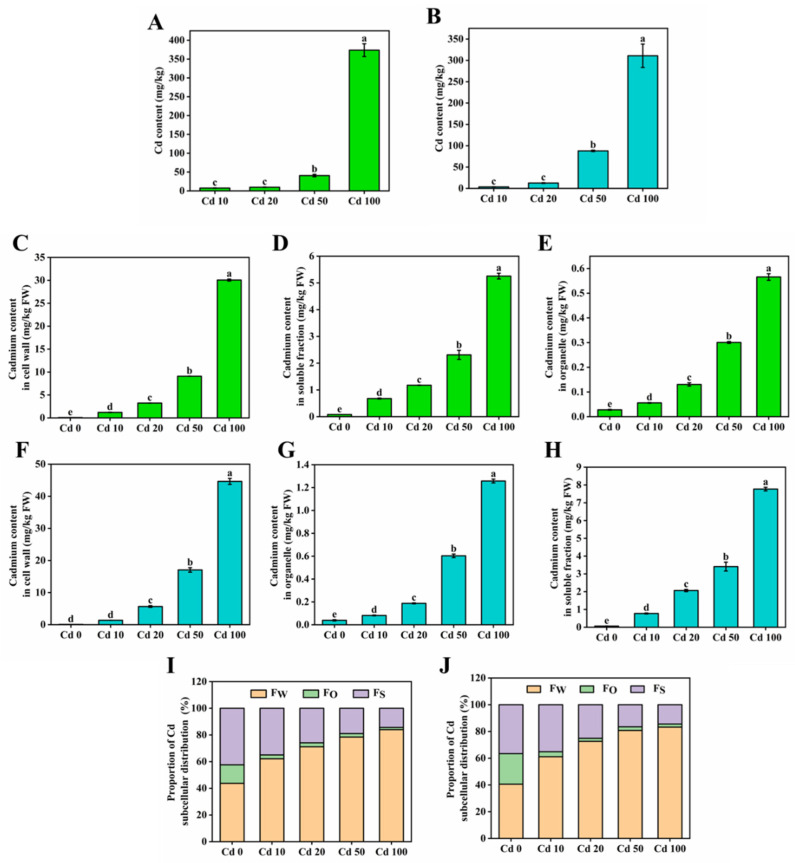
Cd accumulation and subcellular distribution. (**A**,**B**) Cd content in *L. edodes* 1504 (**A**) and L130 (**B**); (**C**–**E**) the Cd content of cell wall (**C**), organelle (**D**), and soluble fractions (**E**) in 1504; (**F**–**H**) the Cd content of cell wall (**C**), organelle (**D**), and soluble fractions (**E**) in L130; (**I**,**J**) the proportions of Cd in different subcellular fractions in 1504 (**I**) and L130 (**J**). F_W_, Cd content in the cell wall; F_O_, Cd content in organelles; F_S_, Cd content in the soluble fraction. Values in a column followed by different letters are significantly different (*p* < 0.05).

**Figure 7 microorganisms-13-00062-f007:**
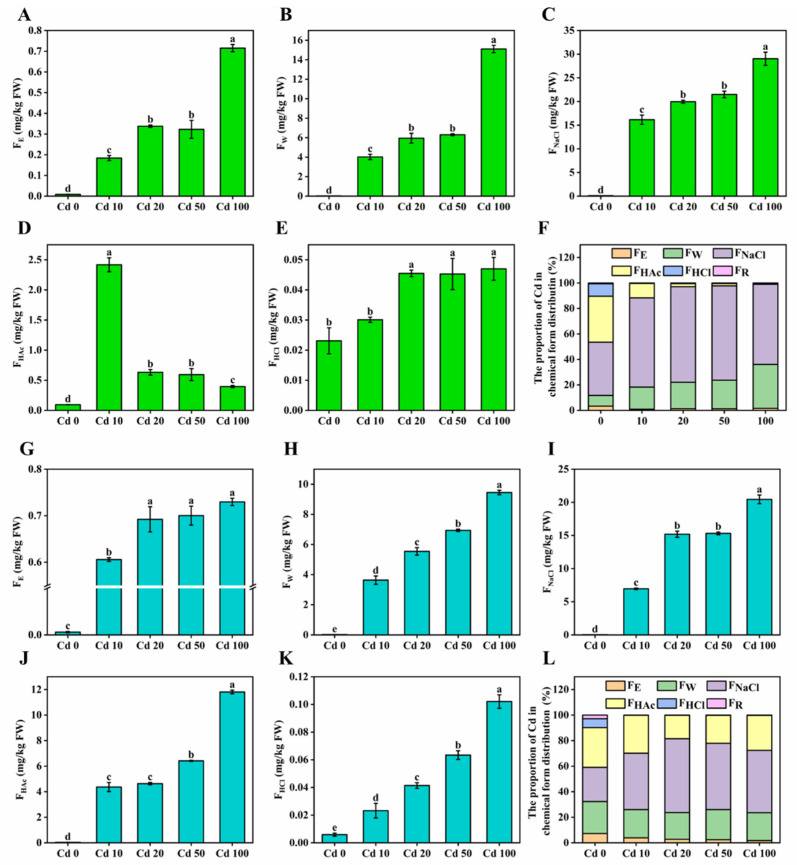
Cd chemical forms. (**A**–**E**) The content of Cd in 80% ethanol- (**A**), H_2_O- (**B**), NaCl- (**C**), Hac- (**D**), and HCl-extractable (**E**) states in 1504; (**F**) the proportions of different chemical forms in 1504; (**G**–**K**) the content of Cd in 80% ethanol- (**G**), H_2_O- (**H**), NaCl- (**I**), Hac- (**J**), and HCl-extractable (**K**) states in L130; (**L**) the proportions of different chemical forms in L130. F_E_, inorganic species extracted by 80% ethanol; F_W_, water-soluble species extracted by deionized water; F_NaCl_, pectate and protein-integrated species extracted by 1 M NaCl; F_HAc_, phosphate-bound species extracted by 2% acetic acid; F_HCl_, oxalate-bound species extracted by 0.6 M HCl; F_R_, Cd in residues. Values in a column followed by different letters are significantly different (*p* < 0.05).

## Data Availability

The original contributions presented in this study are included in the article. Further inquiries can be directed to the corresponding author.

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
