# Peer review of "Cadmium Tolerance and Detoxification Mechanisms of Lentinula edodes: Physiology, Subcellular Distribution, and Chemical Forms"

_microorganisms, 2025, doi:10.3390/microorganisms13010062_

Round 1
Reviewer 1 Report
Comments and Suggestions for Authors
The aim of this study was to investigate the physiological response of Lentinula edodes to different concentrations of cadmium, as well as the subcellular distribution and chemical forms of cadmium. Instrumental and molecular methods were used to evaluate cadmium accumulation and the impact of Cd on fungal mycelium. The study is interesting but requires some improvements:
- The purity and manufacturer of all reagents, including the growing media, should be provided for reproducibility.
- All equations must be numbered sequentially for easy reference.
- As the reference citations are numeric, the year of publication should be removed from the main text to maintain consistency.
- How many repetitions were performed for each experimental setup, including the fungal growth experiments? Please specify.
- It is unclear whether all tests were performed on solid media, or if liquid media was also used. This should be clarified in the methods section.
- In the introduction, it was stated: “Two Lentinula edodes strains with contrasting cadmium tolerance were selected in the present study.” Were any preliminary tests performed to evaluate cadmium tolerance, or is this information based on prior knowledge? Additionally, were there any other differences between the selected strains apart from cadmium tolerance?
- Please specify the type of cadmium salt (e.g., CdClâ‚‚ or other) used in the study. Discuss whether similar results would be expected if a different cadmium salt were used.
- The abbreviation "CK" needs to be explained clearly in the text for readers unfamiliar with the term.
- Figure 6 (A, B): Where does the cadmium in the control samples come from? The absence of "Cd 0" results makes this unclear. While it is understandable that an initial amount of cadmium may naturally exist in the fungi, there should be a logical explanation for why the Cd50 and Cd100 results differ by more than a factor of two. This discrepancy needs to be addressed.
Author Response
1. The purity and manufacturer of all reagents, including the growing media, should be provided for reproducibility.
Response: Thank you for your suggestion. According to your comment, we have added the detailed information about reagents.
2. All equations must be numbered sequentially for easy reference.
Response: Thank you for your suggestion. According to your comment, all equations have been numbered sequentially.
3. As the reference citations are numeric, the year of publication should be removed from the main text to maintain consistency.
Response: Thank you for your suggestion. According to your comment, we have deleted the year of publication in the main text to maintain consistency.
4. How many repetitions were performed for each experimental setup, including the fungal growth experiments? Please specify.
Response: Thank you for your suggestion. All experiments were carried out in triplicate. We have added the relative information in “Materials and Methods” section.
5. It is unclear whether all tests were performed on solid media, or if liquid media was also used. This should be clarified in the methods section.
Response: Thank you for your suggestion. The mycelia were cultivated in solid media with agar, and this has been clarified in the “Materials and Methods” section.
6. In the introduction, it was stated: “Two Lentinula edodes strains with contrasting cadmium tolerance were selected in the present study.” Were any preliminary tests performed to evaluate cadmium tolerance, or is this information based on prior knowledge? Additionally, were there any other differences between the selected strains apart from cadmium tolerance?
Response: Thank you for your suggestion. In our previous research, 10 main cultivated strains from different regions of China were selected to investigate the cadmium tolerance. According to the growth rate and biomass of 10 strains under different cadmium stress, 2 strains with contrasting cadmium tolerance were screened.
7. Please specify the type of cadmium salt (e.g., CdClâ‚‚ or other) used in the study. Discuss whether similar results would be expected if a different cadmium salt were used.
Response: Thank you for your suggestion. The type of cadmium salt was CdClâ‚‚. According to your comment, we have specified the relative information in the “Materials and Methods” section. In our subsequent research, the effects of different cadmium salt (CdClâ‚‚, CdNO3, and CdSO4) on mycelia were investigated. We found that the different types of cadmium salt showed no significant difference on growth and biomass of mycelia. We hope that the relevant research results will be published in the future.
8. The abbreviation "CK" needs to be explained clearly in the text for readers unfamiliar with the term.
Response: Thank you for your suggestion. According to your comment, we have added the detailed information about “CK” in the text.
9. Figure 6 (A, B): Where does the cadmium in the control samples come from? The absence of "Cd 0" results makes this unclear. While it is understandable that an initial amount of cadmium may naturally exist in the fungi, there should be a logical explanation for why the Cd50 and Cd100 results differ by more than a factor of two. This discrepancy needs to be addressed.
Response: Thank you for your suggestion. The Lentinula edodes has strong capacity to enrich cadmium. It is inevitable that the culture medium contains trace amounts of cadmium. Thus, a low content of cadmium was detected in the mycelia using graphite furnace atomic absorption spectrometry. The cadmium accumulation in mycelia treated with Cd100 was 2 times higher than that in Cd50. This may be caused by the following two reasons. (1) Under high cadmium stress, the growth of mycelia was significantly inhibited, resulting in the remarkable decrease in the biomass of collected mycelia, especially for mycelia treated with Cd100. (2) As presented in our research, cadmium concentration could affect the chemical forms of cadmium in mycelia, and higher cadmium treatment induced an increase in the proportion of cadmium with lower toxicity, such as NaCl and HAc extractable cadmium, which resulted in the significant increase in both cell wall and soluble fractions.
Reviewer 2 Report
Comments and Suggestions for Authors
The introduction provides a good overview of many aspects discussed in this work. The aim is clearly defined. The methodology is sufficiently described in 11 sections. In individual experiments, the authors often refer to the appropriate literature. The authors based their experiments on two Lentinula edodes strains (1504 and L130, ) with contrasting cadmium tolerance. The results indicated that appropriate low cadmium stress promoted, while higher cadmium content inhibited the mycelial growth. The effect of different cadmium content on mycelial morphology and cellular ultrastructure was characterized using SEM and TEM. Higher cadmium stress caused damage to the cells by impairing membrane permeability and integrity. In addition, results regarding the effect of cadmium concentration on functional groups and on the antioxidant system were presented. It was shown that cadmium stress could stimulate the production of ROS, which leads to cell damage. The important conclusions include the statement that cadmium stress stimulated the activities of superoxide dismutase and catalase, and these two enzymes participated in the detoxification of cadmium. Furthermore, it was shown that: i) cadmium was dominantly distributed in cell wall fraction and that the share of cadmium in the cell wall fraction increased with the increasing cadmium exposure, ii) the cell wall is the first barrier, which reduces cadmium toxicity and that iii) the NaCl-extracted state was the predominant form of cadmium in the tested strains. The results of the studies are documented in detail using communicative figures separately for the tested strains 1504 and L130. The results presented in detail in this form are very valuable because they allow for understanding various aspects of tolerance and detoxification mechanisms of cadmium in L. edodes. The manuscript is prepared carefully (see Remarks). The conclusions are correct. The manuscript should be published in Microorganisms.
Remarks
Line 49 if abbreviations are used for the first time in the text, they should be explained even if they are commonly known
Line 54 hydroxyl, carboxylic, and amino, - this requires clarification
Line 80 L. edodes strains (L130 and 1504) - if the sequences are deposited in GenBank, the numbers should be given here
Line 88 7.5 is the diameter of the inoculum? - this should be explained
Line 89 consider revising this sentence
Line 278 - this is a repetition of the text from line 270
Line 303 Line 306 it should be Fig. 6, (Fig. 6 C-H)- add a space, also in other places in the text
Author Response
1. Line 49 if abbreviations are used for the first time in the text, they should be explained even if they are commonly known
Response: Thank you for your suggestion. According to your comment, we have added the detailed information about relative enzymes.
2. Line 54 hydroxyl, carboxylic, and amino, - this requires clarification
Response: Thank you for your suggestion. According to your comment, we have further clarified the relevant expressions by carefully reading the previously reported article.
3. Line 80 L. edodes strains (L130 and 1504) - if the sequences are deposited in GenBank, the numbers should be given here
Response: Thank you for your suggestion. The selected two L. edodes strains were provided by the Wuhan Academy of Agricultural Sciences, Wuhan, China, and the sequences are not deposited in GenBank.
4. Line 88 7.5 is the diameter of the inoculum? - this should be explained
Response: Thank you for your suggestion. 7.5 is the diameter of the inoculum, and we have added the detailed information in the text.
5. Line 89 consider revising this sentence
Response: Thank you for your suggestion. According to your comment, we have corrected the inappropriate expression, “D is the diameter of the colony diameter”.
6. Line 278 - this is a repetition of the text from line 270
Response: Thank you for your suggestion. According to your comment, we have deleted the repetitive expressions.
7. Line 303 Line 306 it should be Fig. 6, (Fig. 6 C-H)- add a space, also in other places in the text
Response: Thank you for your suggestion. According to your comment, we have corrected these format errors in the full text.
Reviewer 3 Report
Comments and Suggestions for Authors
Dear authors,
I read with interest the manuscript entitled „Cadmium Tolerance and Detoxification Mechanism of 2 Lentinula edodes: Physiology, Subcellular Distribution and 3 Chemical Forms”, this research presents a significant contribution to understanding cadmium tolerance and detoxification mechanisms in Lentinula edodes, an area that has been relatively unexplored. The study highlights physiological, subcellular, and biochemical responses to cadmium stress in two contrasting strains, providing novel insights into how antioxidant enzyme activity, functional group modulation, and subcellular cadmium distribution mitigate toxicity. The identification of the cell wall as the primary site of cadmium accumulation and the role of specific cadmium chemical forms, such as HAc-extractable cadmium, represents an important advancement in the field. This research is innovative in elucidating strain-specific differences in cadmium handling and presents valuable findings with implications for food safety and environmental management. Its scientific rigor and relevance justify its acceptance for publication. The manuscript is interesting, brings a degree of novelty, well structured, and as such I agree with its publication.
However, I have a question:
Based on the results presented, I would like to inquire whether such methods have already been implemented or if their development is feasible. Specifically, this refers to the enhancement of metal-binding molecule production, such as metallothioneins (proteins) and phytochelatins (small peptides).
Congratulations to the authors
Author Response
1. Based on the results presented, I would like to inquire whether such methods have already been implemented or if their development is feasible. Specifically, this refers to the enhancement of metal-binding molecule production, such as metallothioneins (proteins) and phytochelatins (small peptides).
Response: Thank you for your question. In our previous study, a novel non-metallothionein cadmium-binding protein (LECBP) was isolated from Lentinula edodes, the expression of LECBP in E. coli significantly enhanced the cadmium biosorption capacity of transgenic E. coli. Thus, LECBP could act as a potential remediation tool for cadmium in E. coli.